# Transcriptome Analysis Reveals That C17 Mycosubtilin Antagonizes *Verticillium dahliae* by Interfering with Multiple Functional Pathways of Fungi

**DOI:** 10.3390/biology12040513

**Published:** 2023-03-29

**Authors:** Qi Zhang, Rongrong Lin, Jun Yang, Jingjing Zhao, Haoran Li, Kai Liu, Xiuhua Xue, Huixin Zhao, Shengcheng Han, Heping Zhao

**Affiliations:** 1Beijing Key Laboratory of Gene Resource and Molecular Development, College of Life Sciences, Beijing Normal University, Beijing 100875, China; 2Xinjiang Key Laboratory of Special Species Conservation and Regulatory Biology, College of Life Science, Xinjiang Normal University, Urumqi 830054, China; 3Academy of Plateau Science and Sustainability of the People’s Government of Qinghai Province & Beijing Normal University, Qinghai Normal University, Xining 810008, China

**Keywords:** C17 mycosubtilin, *Verticillium dahliae* 991, transcriptomics, antifungal, *Bacillus subtilis* J15

## Abstract

**Simple Summary:**

*Verticillium dahliae* (Vd) is a kind of soil-borne filamentous fungus that can cause Verticillium wilt in many crops and serious damage to agricultural production. Vd 991 is a main pathogen of cotton Verticillium wilt. Previously, we isolated C17 mycosubtilin from *Bacillus subtilis* J15 (BS J15) broth and showed a significant inhibitory effect on cotton Verticillium wilt. However, the specific fungistatic mechanism by which C17 mycosubtilin antagonizes Vd 991 is not clear. Here, we found that C17 mycosubtilin inhibited the growth and spore germination of Vd 991. C17 mycosubtilin treatment caused shrinking, sinking, and even damage to spores, resulting in rough deformation and uneven distribution of hyphal contents, damage to the cell membrane and cell wall, and mitochondrial swelling of fungi. Moreover, C17 mycosubtilin induced necrosis of Vd 991 in a time-dependent manner. Differential transcription analysis showed that C17 mycosubtilin inhibited fungal growth mainly by destroying the cell structure, blocking the cell cycle, disturbing substance, energy metabolism, and so on. Our results reveal the potential mechanism by which C17 mycosubtilin antagonizes Vd 991, providing useful information for development of more effective antimicrobials.

**Abstract:**

Verticillium wilt is a kind of soil-borne plant fungal disease caused by *Verticillium dahliae* (Vd). Vd 991 is a strong pathogen causing cotton Verticillium wilt. Previously, we isolated a compound from the secondary metabolites of *Bacillus subtilis* J15 (BS J15), which showed a significant control effect on cotton Verticillium wilt and was identified as C17 mycosubtilin. However, the specific fungistatic mechanism by which C17 mycosubtilin antagonizes Vd 991 is not clear. Here, we first showed that C17 mycosubtilin inhibits the growth of Vd 991 and affects germination of spores at the minimum inhibitory concentration (MIC). Morphological observation showed that C17 mycosubtilin treatment caused shrinking, sinking, and even damage to spores; the hyphae became twisted and rough, the surface was sunken, and the contents were unevenly distributed, resulting in thinning and damage to the cell membrane and cell wall and swelling of mitochondria of fungi. Flow cytometry analysis with ANNEXINV-FITC/PI staining showed that C17 mycosubtilin induces necrosis of Vd 991 cells in a time-dependent manner. Differential transcription analysis showed that C17 mycosubtilin at a semi-inhibitory concentration (IC50) treated Vd 991 for 2 and 6 h and inhibited fungal growth mainly by destroying synthesis of the fungal cell membrane and cell wall, inhibiting its DNA replication and transcriptional translation process, blocking its cell cycle, destroying fungal energy and substance metabolism, and disrupting the redox process of fungi. These results directly showed the mechanism by which C17 mycosubtilin antagonizes Vd 991, providing clues for the mechanism of action of lipopeptides and useful information for development of more effective antimicrobials.

## 1. Introduction

Vd is a soil-borne filamentous fungus that can cause Verticillium wilt by invading the xylem of host plants and blocking their vascular bundles [1,2]. It has a wide range of hosts and can infect approximately 660 species of plants belonging to *Solanaceae*, *Leguminosae*, *Cruciferae*, etc. [1,3], and its dormant structure-microsclerotia can survive in soil for more than ten years [1,4]. Therefore, Verticillium wilt caused by Vd is called “plant cancer” and causes serious damage to the world economy [5].

Among the methods to control Verticillium wilt, chemical control (spraying chemical pesticides, soil fumigation, etc.) threatens human health and the ecological environment; progress is difficult regarding breeding resistant varieties due to the lack of highly resistant germplasm and the small target area [4,6]; biological control agents (BCA) have been widely studied because of their low toxicity, low pollution, and high effectiveness [7]. Moreover, many biocontrol strains (50% are *Bacillus*) and BCA have been used to control a variety of fungal diseases [8]. Among them, cyclic lipopeptide (CLP) compounds (including surfactins, fengycins, and iturins) have been widely studied due to their structural and functional diversity. It has been reported that surfactins acted on the cell membrane and exhibited strong haemolytic, anti-inflammatory, anti-tumour, and anti-bacterial activities [9,10]. Fengycin family compounds can damage cell membrane and cell wall of various plant pathogenic bacteria, such as *Mycosphaerella fijiensis* [11], *Fusarium oxysporum* f.sp. *lycopersici* [12], and *Candida albicans* [13], showing broad-spectrum antifungal activity.

Iturins are the largest family of CLPs, including iturins (iturin A, C, D, E) [14,15,16], mycosubtilin [17], bacillomycins (bacillomycin D, L, F) [18,19,20], mojavensin [21], and bacillopeptin [22]. Members of the family exhibit strong antagonistic properties against a variety of phytopathogenic fungi. For a long time, research on this family mainly focused on iturin A. Aranda’s research suggests that iturin A acts as an antifungal by interacting with sterols or ergosterols on target cell membranes to disrupt integrity of cell membranes [23]. Iturin A showed significant inhibitory effects on *Aspergillus carbonarius* [24], *Gaeumannomyces graminis* var. *tritici* [25], *Rhizopus stolonifera* [26], *F. oxysporum*, *Botrytis cinerea* [27], *Phoma tracheiphila* [28], *C. albicans* [29], and *F. graminearum* [30]. Among all these lipopeptides, mycosubtilin is considered the most powerful antifungal lipopeptide [31], but there have been few studies on it. In addition, research on the mode of action of mycosubtilin is mostly conducted by in vitro physical means. For example, biomimetic membranes (Langmuir monolayers and multilayer membranes) have been used to analyse the interactions between mycosubtilin and biological membranes and it was found that mycosubtilin could interact specifically with cholesterol-containing artificial membranes and ergosterol-containing interfacial monolayers [32,33]. It has been found that mycosubtilin could antagonize the pathogens of powdery mildew, downy mildew, and scab, such as *oomycete Bremia lactucae*, *Botrytis cinere*, *F. graminearum*, and *Zymoseptoria tritici* [34,35,36], but the specific fungistatic mechanism is not clear.

In the early stage of the study, we isolated an active substance that can antagonize Vd 991 from the secondary metabolites (SMs) of BS J15 [37]. It had a significant control effect on cotton Verticillium wilt and was identified as C17 mycosubtilin, whose structure was a cyclic heptapeptide (with the amino acid sequence -L-Asn-D-Tyr-D-Asn-L-Gln-L-Pro-D-Ser-L-Asn-) linked to a 17-carbon-atom aliphatic chain by the β-amino group [38]. Vd 991 is a strong pathogenic fungi of cotton Verticillium wilt. There are few reports about the effect of C17 mycosubtilin on prevention and treatment of Verticillium wilt, let alone the antagonistic and cytotoxicity mechanism effect of C17 mycosubtilin on Vd 991. In this study, we first evaluated the effects of C17 mycosubtilin on the growth and spore germination of Vd 991, then explored the toxic phenotype and lethal mode of C17 mycosubtilin to Vd 991 spores and hyphae by microscopy and flow cytometry, and then revealed the lethal mechanism of C17 mycosubtilin to Vd 991 by differential transcription analysis. Finally, the experimental results were verified by quantitative reverse transcription PCR (qRT-PCR). This study fills the gap of the antagonistic effect of C17 mycosubtilin on Vd 991, suggests a new direction for research on the mechanism of lipophytic peptides, and provides effective information for development of more effective antimicrobials.

## 2. Materials and Methods

### 2.1. Strains and Culture Conditions

BS J15 strain used in this study was previously isolated from healthy continuous cropping cotton field soil in Heshuo County, Xinjiang Province, China [39]. The strain frozen at −80 °C was first streaked on beef extract–peptone medium (5 g/L beef extract, 5 g/L sodium chloride, 10 g/L peptone, 15 g/L agar, pH 7) and activated at 37 °C. Then, a single colony was picked, inoculated into liquid beef extract–peptone medium, and cultured at 37 °C with shaking at 200 rpm for subsequent production of C17 mycosubtilin.

Vd 991 was obtained from Xinjiang Academy of Agricultural Sciences [39]. The strain cryopreserved at −80 °C was first streaked on a Czapek–Dox agar medium [40] plate and activated at 25 °C. Then, a single colony was picked and cultured in Czapek–Dox broth medium for 3–5 days with shaking at 25 °C and 180 rpm. Finally, the culture medium was filtered through a 100 μm aseptic filter to remove the hyphae. The spore fluid was collected and counted with a haemocytometer for other experiments.

### 2.2. Purification of C17 Mycosubtilin

C17 mycosubtilin was isolated from the fermentation broth of BS J15 according to the method of Lin [38]. Briefly, BS J15 was cultured in beef extract–peptone medium (0.5% beef extract, 1% peptone, 0.5% NaCl, pH 7.2) at 37 °C for 24 h. Then, the supernatant of the fermentation broth was collected by centrifugation, precipitated by acid (adjusted to pH 2.0 with 6 N HCl), extracted using 80% (*v*/*v*) acetone, and finally separated by a semi-preparative high-performance liquid chromatography system (semi-prep HPLC, C18 column, 5 μm, 250 × 10 mm, Hypersil GOLDTM, CA) after water-saturated n-butanol extraction. Elution was performed with a gradient of 40–50% acetonitrile (0.05% TFA, *v*/*v*) at a flow rate of 2 mL min^−1^ and monitored at 215 nm.

The peak 4 (P4) fraction, which is C17 mycosubtilin [38], was collected and concentrated, and then the inhibitory activity of C17 mycosubtilin against Vd 991 was detected by a paper disc agar diffusion assay [41,42].

### 2.3. Effects of C17 Mycosubtilin on Growth and Spore Germination of Vd 991

Eight microlitres of Vd 991 spores were inoculated into a 24-well plate at a final concentration of 1.0 × 10^6^ cfu/mL, and different concentrations of C17 mycosubtilin were added to each column. Then, the plate was cultured at 25 °C for 72 h at 180 rpm, the growth status of the fungi was observed, and the colony area was measured with ImageJ software (Version 1.51j8). The minimum concentration that inhibited the growth of Vd 991 for 72 h was defined as the MIC. The IC50 was defined as the concentration at which the colony growth area of Vd 991 was half that of the control at 72 h. In this experiment, the blank control (CK) group was only supplemented with Czapek–Dox broth medium but without C17 mycosubtilin and spores; the control group was only supplemented with Czapek–Dox broth medium and spores but without C17 mycosubtilin.

The colony growth rate was used to characterize the effect of C17 mycosubtilin on the growth of Vd 991. The spores were coated on Czapek–Dox agar medium and cultured at 25 °C for 5 days. Then, circular bacterial blocks with a diameter of 8 mm were inoculated in the centre of Czapek–Dox agar plates containing different concentrations of C17 mycosubtilin (0, 1, 3, 10 μg/mL, of which 0 μg/mL was the control group and the other concentrations were the experimental group) and cultured at 25 °C. On the 3rd, 6th, 12th, and 18th days, the colony diameters were measured using ImageJ software (Version 1.51j8) by the criss-cross method [43], and the growth inhibition degree of C17 mycosubtilin to Vd 991 was calculated according to the following formula: inhibition rate (%) = [(colony diameter of control group-colony diameter of experimental group)/colony diameter of control group] × 100%.

The spores of Vd 991 were inoculated into Czapek–Dox broth medium containing different concentrations of C17 mycosubtilin at a final concentration of 1.0 × 10^6^ cfu/mL and cultured at 25 °C for 6, 12, 18, and 24 h. Then, the spores were collected by centrifugation at 13,000 rpm and resuspended in a small amount of medium. At least 200 spores were photographed and counted under inverted fluorescence microscopy (Observer Z1, ZEISS, Oberkochen, Germany), and the spore germination rate was calculated according to the following formula: germination rate (%) = [number of germinated spores/(number of germinated spores + number of ungerminated spores)] × 100%.

### 2.4. Observation of Vd 991 Micromorphology by Scanning Electron Microscopy (SEM) and Transmission Electron Microscopy (TEM)

The effect of C17 mycosubtilin on conidia of Vd 991 was observed by SEM according to the method of Bo et al. [44]. Vd 991 spores (1.0 × 10^6^ cfu/mL) were cultured in Czapek–Dox broth medium containing different concentrations of C17 mycosubtilin (0, 1, 3, 10 μg/mL) for 6, 12, 18, and 24 h. The spores were collected by centrifugation at 12,000 rpm, resuspended in 2.5% (*v*/*v*) glutaraldehyde (Cat#P1126, Solarbio, Beijing, China), fixed at 4 °C for 2 h, and then washed three times with phosphate buffered saline (PBS, pH 7.2). The samples were dehydrated in the order of 30%, 50%, 70%, 85%, 95%, and 100% ethanol concentrations and then dried. Finally, the samples were fixed on a circular stage, gold (thickness of 20 nm) was sputtered using an EMS-550, and they were observed under SEM (JEOL it300, JEOL, Akishima, Japan) at 10 kV.

The method of observing hyphae by SEM was modified from the method of observing spores. A round Vd991 block with a diameter of 8 mm was inoculated on a Czapek–Dox agar plate, and cover slips (thickness of 0.1 mm) were inserted obliquely 45° around the block and cultured at 25 °C for 3 days. The slips were gently removed and transferred to Czapek–Dox broth medium containing different concentrations of C17 mycosubtilin (0, 1, 3, 10 μg/mL) for 6, 12, 18, and 24 h. Then, the slips were fixed in 2.5% (*v*/*v*) glutaraldehyde at 4 °C for 2 h and washed three times with PBS. The subsequent steps of dehydration, drying, and microscopic observation were consistent with those for observing spores.

The sample preparation method for TEM was similar to that for scanning electron microscopy. The collected spores were fixed with 2.5% (*v*/*v*) glutaraldehyde and 2% paraformaldehyde (*v*/*v*) in phosphate buffer (PB) (0.1 M, pH 7.4) and washed twice in PB and twice in ddH_2_O. Then, samples were first immersed in 1% (*wt*/*v*) OsO_4_ and 1.5% (*wt*/*v*) potassium ferricyanide aqueous solution at 4 °C for 2 h. After washing, tissues were dehydrated through graded alcohol (30%, 50%, 70%, 80%, 90%, 100%, 100%, 10 min each) in pure acetone (2 × 10 min). Samples were infiltrated in graded mixtures (8: 1, 5: 1, 3: 1, 1: 1, 1: 3, 1: 5, 1: 8) of acetone and Spurr’s resin (10 g ERL 4221, 8 g DER 736, 25 g NSA, and 0.7% DMAE) and then changed to pure resin. Finally, tissues were embedded in pure resin and polymerized for 12 h at 45 °C and for 48 h at 70 °C. The ultrathin sections (70 nm thick) were sectioned with a microtome (Leica EM UC6), double-stained with uranyl acetate and lead citrate, and examined using a transmission electron microscope (FEI Tecnai Spirit120kV) under 100 kV voltage.

### 2.5. ANNEXIN V-FITC/PI Staining to Detect the Lethal Effect of C17 Mycosubtilin on Conidia of Vd 991

The lethal effect of C17 mycosubtilin on Vd 991 conidia was detected by an ANNEXIN V-FITC/PI Apoptosis Detection Kit (Cat#CA1020, Solarbio, Beijing, China). Vd 991 spores (1.0 × 10^6^ cfu/mL) were cultured at 25 °C for 2, 6, 12, and 24 h in Czapek–Dox broth medium containing various concentrations of C17 mycosubtilin (1, 3, 10 μg/mL). Untreated spores served as controls. Spores (1.0 × 107) were collected by centrifugation, washed twice with precooled PBS, and then resuspended in 1 mL of 1× binding buffer. Then, 100 µL of spore suspension was incubated with 5 µL of Annexin V-FITC and 5 µL of propidium iodide (PI) for 10 min and 5 min in the dark, respectively. Finally, 4 times the volume of PBS was added to terminate the reaction, and then it was detected by flow cytometry (NovoCyte3130, Agilent, Santa Clara, CA, USA).

The experiment of detecting the effect of C17 mycosubtilin on membrane integrity by PI staining was basically consistent with the experimental method of detecting the lethal effect of C17 mycosubtilin on Vd 991 conidia except that only PI was added during staining.

### 2.6. RNA Sequencing and Data Analysis

The conidia of Vd 991 were cultured in Czapek–Dox broth medium with IC50 or without C17 mycosubtilin for 2 h and 6 h at 25 °C. M0H6 and M0H2 indicated that the spores grew naturally in Czapek–Dox broth medium without C17 mycosubtilin treatment for 6 h and 2 h, respectively; M3H6 and M3H2 indicated that the spores grew in Czapek–Dox broth medium containing C17 mycosubtilin at the IC50 concentration for 6 h and 2 h, respectively. The spores were collected by centrifugation (12,000 rpm, 4 °C, 10 min), washed once with PBS, and then immediately frozen in liquid nitrogen for RNA extraction. Total RNA was extracted from the samples using TRIzolTM Reagent (Invitrogen Life Technologies, Waltham, MA, USA) according to the manufacturer’s instructions. The integrity of RNA was determined by electrophoresis, and the concentration of RNA was detected based on the A260/A280 absorbance ratio with a Nanodrop ND-2000 system (Thermo Scientific, Waltham, MA, USA).

High-quality RNA samples were subsequently used to construct cDNA libraries and sequencing on the Illumina NovaSeq 6000 Platform (Shanghai Application Protein Technology Co., Ltd., Shanghai, China). Clean data were obtained from the raw data by removing the adapter sequence and filtering out low-quality and multi-N reads (N means that the base information cannot be determined). Then, clean reads were separately aligned to the genome sequences of *Verticillium dahliae* VdLs.17 by using HISAT2 software [45]. Full sequences and annotations were downloaded from the NCBI Genome (https://www.ncbi.nlm.nih.gov/genome/?term=txid27337[orgn], accessed on 31 August 2022). Feature Counts software (http://subread.sourceforge.net/, accessed on 31 August 2022) was used to count the read numbers mapped to each gene and then calculate the FPKM value of each gene in each sample. Differential expression analysis was performed by DESeq2 [46]. Genes with a Padj (adjusted *p*-value) <0.05 and absolute fold change ≥2 were considered significantly differentially expressed genes (DEGs). Kyoto Encyclopaedia of Genes and Genomes (KEGG) pathway enrichment analysis of DEGs was performed using the cluster-Profiler R package [47,48]. When FDR (*p*-value corrected by Benjamini and Hochberg method) <0.05, the pathway was considered significantly enriched. Gene ontology (GO) function enrichment analysis was performed using the OmicStudio tools (https://www.omicstudio.cn/tool, accessed on 31 August 2022). Go-terms with a *p*-value < 0.05 were considered significantly enriched [49].

RNA-Seq data were deposited to the NCBI Sequence Read Archive with registration number PRJNA868877.

### 2.7. qRT–PCR

Expression analysis of genes was detected by qRT–PCR. Total RNA was extracted from samples that were treated as described for RNA sequencing. Approximately 2 μg of RNA was reverse-transcribed using a First-Strand cDNA Synthesis Super Mix Kit (Trans, Shanghai, China) and used as a template for qRT–PCR. qRT–PCR was performed using an ABI Quant Studio 6 Flex Real-Time PCR System (Applied Biosystems, San Francisco, CA, USA) with Power SYBR Green PCR Master Mix (Applied Biosystems, San Francisco, CA, USA). The thermal program was 10 min at 95 °C, followed by 40 cycles of 15 s at 95 °C and 60 s at 60 °C. The dissociation curve program was used to confirm the specificity of the target amplification product, and three biological replicates were performed per sample. The beta-tubulin gene of Vd (DQ266153) was used as an internal control, and the expression levels were calculated using the CT values and the 2^−ΔΔCT^ method [50]. The primers for the target genes are listed in Appendix A.

### 2.8. Statistical Analysis

Statistical analyses were performed using GraphPad Prism 9.0 software. One-way analysis of variance (ANOVA) and unpaired Student’s *t*-test were used to assess significant differences, where ns indicates no significant difference and *** and **** indicate significant differences at *p* < 0.001 and *p* < 0.0001, respectively.

## 3. Results and Discussion

### 3.1. C17 Mycosubtilin Inhibits the Growth and Conidial Germination of Vd 991 and Affects Its Microscopic Morphology and Organelle Morphological Structure

C17 mycosubtilin strongly inhibited elongation of hyphae and germination of conidia of Vd 991. For Vd 991 spores at a concentration of 1 × 10^6^ cfu/mL, the MIC and IC50 were 3 μg/mL and 1 μg/mL, respectively (Appendix A). As shown in Figure 1A, C17 mycosubtilin inhibited the growth of Vd 991 on the Czapek–Dox plate but significantly inhibited the germination of conidia only at concentrations above 3 μg/mL.

SEM results showed that C17 mycosubtilin significantly changed the morphology and even the survival status of Vd 991 spores and hyphae. When Vd 991 spores were treated with C17 mycosubtilin at MIC, fungal spores could not grow normally to form hyphae, resulting in cytoplasmic pyknosis, surface depression, and even damage and perforation, which was aggravated with extension of treatment time (Figure 1D). At the IC50 concentration, some spores were still able to form hyphae and grow, but, with prolonged treatment time, the fungal spores showed shrinkage and depression and even broke (Figure 1C). Compared with the control group, when treated with MIC, the hyphae became twisted and rough and could not form a sporogenic structure, the surface of the mycelium was sunken, the internal contents were unevenly distributed, and the hyphae gradually became transparent (Appendix A). Under the IC50 treatment concentration, the smooth and full hyphal surface became rough, multiple depressions appeared, and the sporulation ability decreased significantly (Appendix A). Previous studies have shown that iturins could inhibit growth of fungi; fengycin inhibited the growth of *M. fructicola mycelium* [51] and iturin A seriously affected the micromorphology of *F. graminearum*, including mycelium distortion, cell membrane leakage, and plasma membrane separation from the cell wall [30]. Our results suggested that all these changes in appearance may be caused by destruction of the internal cellular structure, which is similar to previous findings [52,53,54].

TEM observations showed that C17 mycosubtilin could damage the cell membrane and cell wall of Vd spores and lead to whole mitochondria swelling. When treated with C17 mycosubtilin at the IC50 concentration, compared with the control group, the mitochondria of the spores in the treatment group were swollen, and, with extension of treatment time, the cell wall and cell membrane became structurally unclear and damaged (Figure 2). Under the treatment concentration of MIC, the spores in the treatment group showed cytoplasm extravasation in addition to whole mitochondrial swelling, internal ridge widening, and damage to the cell wall and membrane (Appendix A). These results were in accordance with findings reported by Wang and colleagues [55], who showed that iturin A caused cell membrane disruption and an irregular internal cell structure in *P. infestans* [55]. In summary, through phenotypic observation, we found that C17 mycosubtilin treatment could affect the external and internal cellular structure of Vd spores and hyphae and had a certain lethal effect on spores.

### 3.2. C17 Mycosubtilin Induces Conidial Necrosis of Vd 991

Annexin V-FITC/PI staining was used to detect and analyse the lethal effect of C17 mycosubtilin on Vd 991 conidia. Vd 991 spores were treated with concentrations of the C17 mycosubtilin homologue of MIC and IC50, respectively. With extension of treatment time, the flow density map showed that the sporophyte gradually migrated from the third quadrant to the second quadrant (Figure 3A and Appendix A). After 24 h of treatment with C17 mycosubtilin at IC50 and MIC concentrations, the proportion of active spores decreased from 97.2% to 59.8% and 51.8%, while the proportion of necrotic spores increased from 1.6% to 34.1% and 41.2%, respectively (Figure 3A and Appendix A). These results indicate that C17 mycosubtilin mainly caused necrotic spores of Vd. Studies on other lipopeptides have shown that iturin A treatment could cause apoptosis of *A. carbonarius*, *P. infestans,* and Caco-2 cells [24,55,56]. Fengycin can also induce apoptosis of *Rhizopus creeping* cells. These results suggest that the antifungal mechanism of C17 mycosubtilin is different from that of other lipopeptides and is thus worthy of more in-depth study. Moreover, PI staining showed the degree of damage to fungal cell membranes [57]. When Vd 991 spores were stained with PI alone, the percentage of fungal spores with membrane damage gradually increased with extension of C17 mycosubtilin treatment time (Figure 3B and Appendix A), which was consistent with the previous electron microscopic observation of sunken damage on the surface of spores and thinning of cell walls.

### 3.3. Overall Effects of C17 Mycosubtilin on the Gene Transcription of Vd 991

To further analyse the lethal mechanism of C17 mycosubtilin on Vd, differential transcriptome analysis was performed and the gene expression profiles of Vd 991 spores treated with C17 mycosubtilin at IC50 concentrations for 2 h and 6 h were studied. The Pearson correlation between samples is an important index to test reliability of experiment and rationality of sample selection. In this study, the correlation coefficients (R) between samples in the same group were all greater than 0.88 (Appendix A), indicating that the similarity of expression patterns between samples was very high and the experimental reproducibility was good. Principal component analysis (PCA) can evaluate differences between groups and quality of biological repetition within groups. The PCA results of this study showed that the samples in each group were clustered within the group and dispersed among the groups (Appendix A), indicating that C17 mycosubtilin treatment was effective, and the expression profile changes caused by different treatment times were different, which ensures the reliability of subsequent transcriptome analysis.

According to the differential expression analysis of different treatments, the genes with twice the fold change and corrected *p* < 0.05 were significantly DEGs, which were represented by a volcano map (Figure 4A–C). As shown in Figure 4E, 457 DEGs (including 227 downregulated genes and 230 upregulated genes) were identified when Vd 991 spores grew in the natural state for 2 h to 6 h. After 2 h of C17 mycosubtilin treatment, 3292 DEGs (including 1367 downregulated genes and 1925 upregulated genes) were identified, including 286 of 457 DEGs identified by natural growth. After 6 h of C17 mycosubtilin treatment, 3112 DEGs (including 1343 downregulated genes and 1769 upregulated genes) were identified, including 185 of 457 DEGs identified by natural growth. Excluding the DEGs obtained in the natural growth state, 3780 C17 mycosubtilin regulatory genes were identified, which were further divided into the following three groups: 853 2–h specific C17 mycosubtilin response DEGs, 774 6–h specific C17 mycosubtilin response DEGs, and 2153 2–h and 6–h common C17 mycosubtilin response DEGs (Figure 4F).

### 3.4. C17 Mycosubtilin Treatment Interferes with Normal Function of Multiple Functional Units and Metabolic Pathways of Vd

Then, all DEGs were enriched by GO functional annotation and KEGG (Kyoto Encyclopaedia of Genes and Genomes) and the characteristics of these three groups of C17 mycosubtilin regulatory genes were analysed. GO annotation analysis showed that 2–h and 6–h common C17 mycosubtilin response DEGs were mainly enriched in DNA-replication-related items and mitotic-cell-cycle-related process items, including chromosome organization, mitotic nuclear division, mitotic sister chromatid segregation, the chromosome part, and the nuclear chromosome part, in the biological process (BP) and cell component (CC) categories (Figure 5A). The molecular function (MF) category was mainly enriched in hydrolase-activity- and oxidoreductase-activity-related items (Figure 5A). The 2–h specific C17 mycosubtilin response DEGs were mainly enriched in cytoskeleton-related terms in the BP and CC categories, including 15 items, such as regulation of supramolecular fibre organization, actin polymerisation or depolymerisation, regulation of actin filament organization, regulation of protein polymerization, Arp2/3 protein complex, and cortical cytoskeleton (Figure 6A). The MF category was mainly enriched in hydrolase-activity- and DNA-replication-related terms (Figure 6A). The 6–h specific C17 mycosubtilin response DEGs were mainly enriched in two types of functions. One was translation-related functions, including 19 items, such as the peptide biosynthetic process, regulation of the cellular amide metabolic process, translational initiation and regulation of translation in the BP section, RNA binding and translation initiation factor activity in the MF section, the eukaryotic translation initiation factor 3 complex, translation preinitiation complex, and ribonucleoprotein complex in the CC section (Figure 7A). The second was transmembrane transport function, including nine items, such as metal ion transmembrane transporter activity, copper ion transmembrane transport, and other terms (Figure 7A).

KEGG analysis showed that 2–h and 6–h common DEGs were mainly related to cell cycle–yeast, glyoxylate, and dicarboxylate metabolism, valine, leucine, and isoleucine degradation, fatty acid metabolism, propionate metabolism, glycolysis/gluconeogenesis, pyruvate metabolism, starch and sucrose metabolism, and fatty acid degradation pathways (Figure 5B). The 2–h specific DEGs mainly played a key role in peroxisome, DNA replication, sphingolipid metabolism, fructose and mannose metabolism, nucleotide excision repair, regulation of actin cytoskeleton, pentose and glucuronate interconversions, and fatty acid biosynthesis pathways (Figure 6B). The 6–h specific DEGs were mainly concentrated in the ribosome, biosynthesis of amino acids, cysteine, and methionine metabolism, 2-oxocarboxylic acid metabolism, phenylalanine, tyrosine and tryptophan biosynthesis, RNA transport, and the pentose phosphate pathway (Figure 7B). In summary, we speculate that C17 mycosubtilin may inhibit growth of fungi by affecting multiple functional pathways.

### 3.5. C17 Mycosubtilin Treatment Disrupts the Structure and Function of Cell Walls and Plasma Membrane of Vd

Integrity of plasma membrane and cell wall is important for function of fungi. Transcriptome analysis showed that, after C17 mycosubtilin treatment, some 2–h and 6–h common DEGs were enriched in membrane-component-related items (intrinsic component of membrane GO:0031224 and integral component of membrane GO:0016021) to regulate formation and function of cell membrane and cell wall (Appendix A Sheet 1). Further functional annotation analysis showed that the genes encoding chitin synthase (VDAG_10179) and cellulose synthase (VDAG_02123) were downregulated, while the genes encoding glycogen debranching enzyme (VDAG_02218), endoglucanase (VDAG_05955, VDAG_02711), and alpha-glucoside permease (VDAG_08286, VDAG_06994) were upregulated (Appendix A and Appendix A, Sheet 1). The fungal cell wall is mainly composed of chitin, glucan, and cellulose, and its structural integrity mainly depends on chitin [24]. A previous study showed that C17 mycosubtilin caused destruction and deformation of plasma membranes and cell walls in *Fusarium graminearum* hyphae [58]. Our results showed that synthesis of important polysaccharides in the fungal cell wall, such as chitin, glucan, and cellulose, was inhibited, and decomposition of these polysaccharides was intensified. It was suggested that the fungal cell wall cannot be formed normally after C17 mycosubtilin treatment. The cell wall is a necessary dynamic organelle for fungi to construct and maintain cell morphology, resist adverse environmental damage, and provide interaction recognition sites, and it is also a potential molecular target in antifungal therapy [59]. After C17 mycosubtilin treatment, glucan 1,3-beta-glucosidase (VDAG_01579), cell wall surface anchor signal protein (GPI-anchored serine-rich protein, VDAG_09779), and other enzymes that have an important impact on cell wall rearrangement, reinforcement, cell recognition, and production of signalling molecules [57,60,61] were found to be downregulated, indicating that integrity of cell wall morphology and function were damaged, which may lead to cell death.

Moreover, some 2–h and 6–h common DEGs enriched in membrane-component-related items associated with organelle integrity and normal functioning, such as integral membrane protein (VDAG_06228, VDAG_02900), plasma membrane ATPase (VDAG_03948, VDAG_03949), and ion transporters (VDAG_06616, VDAG_03807), were downregulated (Appendix A Sheet 1). Expression of genes related to transport of some important intracellular substances and basic energy metabolism, such as SOCE-associated regulatory factor of calcium homoeostasis (VDAG_09767), maltose permease (VDAG_ 01684), lactose permease (VDAG_ 08212), potassium channel (VDAG_05473), and ion-transport-related ATP enzyme (sodium transport ATPase, VDAG_06663; sodium transport ATPase, VDAG_09836), were upregulated (Appendix A Sheet 1). The fungal cell membrane is mainly composed of sterols, glycerol phospholipids, and sphingolipids, which undertake transmembrane transport function of many substances and are the osmotic barrier of small molecules and signal transduction pathways. The results showed that C17 mycosubtilin treatment destroyed the integrity of the fungal membrane system, and expression of some energy-related genes was upregulated due to maintenance of basic physiological activities of cells. This is similar to the previously reported lipopeptide Iturins produced by *Bacillus* by destroying integrity of organelles and plasma membranes to exert antifungal activity [30,62]. We also found that the genes related to exocytosis and secretion (GO:0032940, GO:0046903) of the membrane were downregulated after C17 mycosubtilin treatment, indicating that the vesicle transport capacity of fungi decreased. These results were consistent with the mechanism by which iturin A inhibited growth of *Aspergillus carbonarius* by interfering with its transport and metabolism [24].

In addition, twenty genes (including seven downregulated genes and thirteen upregulated genes) were enriched in terms related to transmembrane transporter activity (GO:0022857) in 6–h specific C17 mycosubtilin response DEGs (Figure 7A and Appendix A and Appendix A Sheet 1). The downregulated genes were mainly related to calcium, zinc, and other important signal ions and enzyme activity centre ions, as well as some amino acid transport. Among them, VDAG_09301 and VDAG_10408 encode calcium-transporting ATPase and zinc-regulated transporter 2, respectively, which are responsible for maintaining intracellular ion concentration gradients. VDAG_06062, encoding dicarboxylic amino acid permease, mediates high-affinity and high-capacity transport of L-glutamate and L-aspartic acid and is a transporter of Gln, Asn, Ser, Ala, and Gly [63,64]. Of note, calcium ions are important signalling molecules that are necessary for growth and reproduction of fungi, and they are components of many cellular structures [65,66]. Zinc is the functional component of many fungal enzymes, such as aldolase [67], alcohol dehydrogenase [68], RNA polymerase aminopeptidase [69], nuclease P1 [70], superoxide dismutase [71], and pyruvate decarboxylase. Amino acid transport is closely related to energy metabolism, signal transduction, and synthesis of macromolecular substances, such as sugars, lipids, and proteins. Downregulation of their expression may cause disorders of the cellular enzyme system and energy and substance metabolism pathways and affect growth and development of fungi. Among the upregulated genes, four genes encoding copper ion transmembrane transporters (VDAG_01771, VDAG_06704, VDAG_06151, VDAG_04029) were located on the tonoplast or plasma membrane. Copper is a basic trace element and in a variety of antioxidant enzymes in fungi, including tyrosinase [72], ascorbate oxidase [73], uridine nucleosidase, and superoxide dismutase [74,75,76]. Copper also participated in antioxidation of cytochrome C and promoted growth and development of the body. Upregulation of these genes suggested that C17 mycosubtilin treatment destroyed the original ion concentration gradient in cells, and cells must maintain the ion concentration gradient as much as possible by enhancing active transport, then activating the activities of related enzymes and eliminating the toxic effects of oxygen free radicals in the body to maintain cell growth and material transport.

Overall, the study showed that C17 mycosubtilin could damage the membrane components and cell walls and affect substance transport and metabolism of fungi, as well as the normal intracellular electrochemical gradient and ion concentration gradient, which was consistent with the results of spore depression and cell damage found via phenotypic observation, resulting in a disorder of the intracellular enzyme system, weakening the energy supply and material transport and absorption capacity of spores, and even causing spore death.

### 3.6. C17 Mycosubtilin Treatment Inhibits DNA Replication of Vd and Blocks Its Cell Cycle Process

DNA replication is the basis of biological heredity and the basis for species to maintain their uniqueness and genetic integrity [77]. GO enrichment analysis showed that C17 mycosubtilin treatment had a significant effect on DNA replication (GO:0006260), especially on DNA-dependent DNA replication (GO:0006261) (Figure 5A). Further analysis showed that expression of 24 genes enriched in the process of DNA replication was downregulated, except for VDAG_05720 and VDAG_10029 (Appendix A and Appendix A Sheet 2). Downregulated genes were involved in DNA replication initiation recognition activity (VDAG_01310), DNA topoisomerase activity (VDAG_06518), DNA helicase activity (VDAG_01493), and DNA polymerase activity (VDAG_06652, VDAG_08342). Among them, DNA polymerase alpha encoded by VDAG_06652 was the only DNA polymerase in eukaryotic cells that could synthesize DNA from scratch. Its main function is to provide primers for DNA replication and play a key role in initiating DNA synthesis. DNA polymerase delta encoded by VDAG_08342 is the main replicase of DNA replication in eukaryotic cells. It is also involved in DNA recombination, damage repair, and cell cycle regulation [78]. Downregulation of these genes directly limited or even blocked the process of DNA replication in fungi, which had a serious impact on growth of fungi. The two upregulated genes mainly played a role in the process of DNA strand break repair. VDAG_05720 encodes DNA ligase 4, which catalyses formation of phosphodiester bonds to connect DNA fragments and acts on VJ recombination and DNA double-strand break repair processes through nonhomologous terminal connections in the human body. VDAG_10029 encodes PPS1 bispecific phosphatase, which mainly plays a role in the DNA synthesis stage of the cell cycle. Overexpression of PPS1 protein in *Saccharomyces cerevisiae* leads to synchronous growth arrest and abnormal DNA synthesis [79]. These results indicated that C17 mycosubtilin treatment inhibited replication of DNA and caused abnormal DNA synthesis; on the other hand, it may cause DNA damage, thus inducing necrosis of spores.

In addition, C17 mycosubtilin treatment also had a significant effect on cell cycle progression of fungi. After C17 mycosubtilin treatment, many genes were enriched in mitotic-cell-cycle-related processes, including 12 items, such as chromosome organization (GO:0051276), mitotic nuclear division (GO:0140014), sister chromatid separation (GO:0000819), chromosome (GO:0005694), and nuclear chromosome (GO:0000228), with a total of 77 genes (Figure 5A and Appendix A Sheet 3). Except for VDAG_03181 and VDAG_07987, the expression of these genes was significantly downregulated (Appendix A), indicating that the cell cycle process of fungi was slowed because of C17 mycosubtilin treatment. Further analysis showed that these downregulated genes were mainly concentrated in the S, G2, and M phases of the cell cycle, involving serine/threonine-protein kinase (VDAG_01807, VDAG_03056), DNA helicase activity (VDAG_01559, VDAG_01493), DNA polymerase activity (VDAG_06652), DNA repair and recombination activity (VDAG_10415), kinetochore protein (VDAG_08677, VDAG_07917), and Rho GTP enzyme activity (VDAG_02011). These enzymes mainly function in key steps, such as DNA replication initiation and cycle transition, chromosome structure maintenance, and spindle checkpoint. Downregulation of their expression will directly delay or even block the cell cycle of fungi and lead to their death.

Studies in animal cells have found that iturin treatment could cause DNA damage and apoptosis in K562 myeloid leukaemia cells, and iturin A could significantly inhibit proliferation of breast cancer cell lines MDA-MB-231 and MCF-7, increase the number of Sub G1 population cells, lead to cell cycle arrest, and induce apoptosis [80,81]. The study in this article has shown that C17 mycosubtilin may affect growth of Vd 991 by inhibiting DNA replication and blocking the cell cycle, which has much in common with the effects of lipopeptides on animal cells. However, our study found that C17 mycosubtilin mainly causes necrotizing death of fungi, while iturins mainly cause apoptosis of animal cells. Therefore, the differences between these two lethal mechanisms need to be further studied.

### 3.7. C17 Mycosubtilin Treatment Inhibits the Translation Process of Vd

Figure 7A shows that C17 mycosubtilin treatment affected the translation process of fungi. Among the 6–h specific C17 mycosubtilin response DEGs, 33 genes were enriched in 19 items related to translation (GO:0006412), and all but VDAG_05416 were downregulated (Appendix A and Appendix A, Sheet 2). Most of the downregulated genes were related to the large subunit (VDAG_07051, VDAG_01336, VDAG_06306, VDAG_04294, VDAG_04766, VDAG_06682) and small subunit (VDAG_07089, VDAG_07517, VDAG_06378, VDAG_08928) of ribosomes. VDAG_06378 encodes 40s ribosomal protein S3 (RPS3), and its sequence domain rearrangement promotes ribosomal 40s subunit maturation [82]. Further, 40s ribosomal protein S0 (RPS0) encoded by VDAG_07089 was necessary for assembly of 40s ribosomal subunits, which was responsible for transforming the 20s rRNA precursor into mature 18s rRNA and maintaining its stability in the later stage of assembly [83]. VDAG_06306 encodes 60s ribosomal protein L25 (RPL25). Its N-terminal nuclear input domain, central RNA binding domain, and C-terminal 60s subunit assembly domain are all necessary for pre-rRNA processing in *Saccharomyces cerevisiae* e [84]. VDAG_01336 encodes 60s ribosomal protein L4 (RPL4), and its conserved internal loop buried deep into the rRNA core is essential for assembly of large ribosomal subunits [85]. In addition, some downregulated genes were related to the translation initiation process (VDAG_00435, VDAG_01482, VDAG_03213, VDAG_10306, VDAG_04458, VDAG_09267, VDAG_02249). The downregulation of their expression directly hindered fungal translation and protein synthesis. This means that C17 mycosubtilin may downregulate fungal protein synthesis by inhibiting ribosomal synthesis, assembly, and translation initiation of Vd spores, resulting in death of spores.

### 3.8. C17 Mycosubtilin Treatment Affects the Hydrolase Activity of Vd

GO enrichment analysis showed that 2–h and 6–h common C17 mycosubtilin response DEGs were mainly enriched in hydrolase activity (GO:0016798) (including hydrolase activity, hydrolysing O-glycosyl compounds (GO:0004553), and hydrolase activity, acting on glycosyl bonds (GO:0016798)) and oxidoreductase activity (GO:0016491) terms in the MF category (Figure 7A and Appendix A, Sheet 4). Among the 45 genes enriched in hydrolase activity, 15 genes were downregulated and 30 genes were upregulated (Appendix A). VDAG_04342 (coding mixed-linked glucanase) and VDAG_08741 (encoding endochitinase) in downregulated genes played a role in glucan synthesis and chitin decomposition, respectively. Treatment of fungi with endochitinase can inhibit fungal spore germination and various fungal pathogen (e.g., *Rhizopus oligosporous*, *Candida albicans,* and *Trichoderma hazianum*) hyphal elongation in vitro [39]. Changes in expression of these two hydrolases may disrupt the integrity of the fungal cell wall. The Gram-negative bacteria-binding protein encoded by VDAG_02722, which belongs to the β-1,3-glucan binding protein family, is the anchoring component of the plasma membrane and is related to recognition of cell surface signal molecules [86]. VDAG_02787 encodes *CRR1p* (belonging to transglycosidase), which is needed for correct assembly of the spore wall. In *Saccharomyces cerevisiae*, *CRR1* mutation made spores sensitive to hydrolase and heat shock treatment, and overexpression of *CRR1* made spores more resistant to adverse conditions, indicating that this gene plays an important role in the correct formation and assembly of the ascospore wall [87]. Downregulation of these genes may lead to a decrease in fungal recognition and resistance to external stimuli, which is not conducive to survival of Vd 991. Additionally, many glycosidases were found in the upregulated genes, such as beta-glucosidase (VDAG_06333, VDAG_05580), alpha-glucosidase (VDAG_01555), beta-galactosidase (VDAG_05347, VDAG_02743, VDAG_03906), beta-mannosidase (VDAG_05563), and beta-xylosidase (VDAG_09302), as well as endoglucanase (VDAG_00931, VDAG_04022) and glucoamylase (VDAG_00408). These genes played an important role in energy release and decomposition of cell wall polysaccharides (such as glucan and cellulose).

Furthermore, among the 2–h specific C17 mycosubtilin response DEGs, 85 genes were enriched in hydrolase activity entries (Figure 6A, Appendix A and Appendix A, Sheet 1). Among them, VDAG_08105, VDAG_02082, and VDAG_03333 were downregulated genes encoding cell wall glycosyl hydrolase, beta-glucanase, and chitin deacetylase, respectively, and genes encoding endoglucanase, exoglucanase, xylanase, and mannosidase were annotated in the upregulated DEGs. Changes in their expression patterns directly affect the structural and functional integrity of the cell wall [88]. Moreover, several ion-transporting ATPases (VDAG_07309) and direct ATP transporters (VDAG_00514, ATP-binding cassette transporter abc1) were annotated in the upregulated genes, suggesting that C17 mycosubtilin treatment may damage energy metabolism of spores. Taken together, the analysis of genes enriched in hydrolase activity further demonstrated that C17 mycosubtilin treatment caused serious damage to fungal cell wall integrity and cell energy metabolism.

### 3.9. C17 Mycosubtilin Treatment Affects the Oxidoreductase System of Vd and Destroys the Process of Material Conversion and Energy Metabolism of Fungi

In this study, GO enrichment analysis showed that C17 mycosubtilin influenced fungal oxidoreductase activity (GO:0016491), especially dioxygenase activity (GO:0051213) and oxidoreductase activity, acting on single donors with incorporation of molecular oxygen (GO:0016701) (Figure 5A). Among them, most of the genes related to dioxygenase activity (GO:0051213) were upregulated, which act on the oxidative metabolism of some aromatic compounds, carotenoids, and keto-aldehydes and had important effects on stress resistance and antioxidation of fungi (Appendix A and Appendix A Sheet 5) [89,90,91]. There are also genes associated with cellular respiration and electron transport chains (VDAG_09687, external NADH-ubiquinone oxidoreductase; VDAG_02096, succinate dehydrogenase flavoprotein; VDAG_00910, cytochrome c), TCA cycles (VDAG_01642, pyruvate dehydrogenase; VDAG_01116, isocitrate dehydrogenase; VDAG_02096, dehydrogenase flavoprotein), fatty acid oxidative metabolism (VDAG_03968, l long-chain specific acyl-CoA dehydrogenase; VDAG_09867 and VDAG_03899, acyl-CoA dehydrogenase; VDAG_08517, 3-hydroxybutyryl-CoA dehydrogenase), and biofilm lipid formation (C-5 sterol desaturase VDAG_00248). Fatty acid desaturase (VDAG_09431;24- dehydrocholsterol reductase VDAG_10090) and other processes are related to amino acid metabolism (Appendix A and Appendix A Sheet 5). Expression of the above genes was downregulated except for genes related to fatty acid oxidation metabolism.

The mitochondrial electron transport chain mainly provides energy for cell survival. External NADH-ubiquinone oxidoreductase encoded by VDAG_09687 is the first enzyme at the entry point of the respiratory chain, and its downregulation leads to a decrease in respiratory energy release of fungi, which is a threat to cell survival. The tricarboxylic acid cycle is not only the main way for the body to obtain energy but also the hub of metabolism of the three major nutrients in the body. Isocitrate dehydrogenase encoded by VDAG_01116 is the rate-limiting step of the TCA cycle, and its downregulation seriously affects the energy supply and material metabolism of the body. These results suggested that C17 mycosubtilin may disrupt the process of intracellular material conversion and energy metabolism by disturbing the redox potential balance and oxidoreductase activity of fungi and then affect the survival state of fungi. Other studies have shown that some lipopeptide antifungal compounds, such as iturin A, and antifungal agents, such as heptanal and fluconazole, exert antifungal effects by disrupting the redox process of fungi [55,57,92]. Our results suggest that C17 mycosubtilin may kill Vd 991 in the same way.

### 3.10. C17 Mycosubtilin Treatment Disrupts Energy Metabolism of Vd and Inhibits Energy Production of Fungal Cells

As shown by KEGG enrichment, 2–h specific DEGs were enriched in fructose and mannose metabolism, pentose and glucuronate interconversions, and the fatty acid biosynthesis pathway (Figure 6B), and 6–h specific DEGs were enriched in 2-oxocarboxylic acid metabolism and the pentose phosphate pathway (Figure 7B). The common DEGs at 2–h and 6–h were enriched in glyoxylate and dicarboxylate metabolism, fatty acid metabolism, glycolysis/gluconeogenesis, pyruvate metabolism, starch and sucrose metabolism, and fatty acid degradation (Figure 5B). These three genes were all enriched in pathways related to energy metabolism, and transmission electron microscopy showed that the whole mitochondrion swelled and the internal ridges became sparse and widened after C17 mycosubtilin treatment. Mitochondrial damage can reduce cell productivity, lead to oxidative stress, and activate the cell death pathway [24,93]. We speculated that C17 mycosubtilin treatment significantly affected the process of energy metabolism of fungi and may reduce energy production of cells.

Detailed analysis showed that the genes related to the energy metabolism pathway were basically upregulated after treatment for 2 h, such as dihydroxyacetone kinase (VDAG_09648) and mannose-6-phosphate isomerase (VDAG_02608), which are related to glycolysis in the fructose and mannose pathways (Appendix A, Sheet 2). In addition, enrichment of 2–h specific DEGs in cytoskeleton-related items played a role in promoting initiation of autophagy, indicating that clearance mechanisms of Vd 991 spores were disrupted and there was accumulation of necrosis that could disrupt cellular homeostasis [94]. These results suggested that, at the initial stage of C17 mycosubtilin treatment, cells may initiate the autophagy pathway and upregulate energy metabolism genes to maintain normal physiological activities to increase cell productivity to address intracellular energy and material metabolism disorders. Furthermore, in a pentose phosphate pathway enriched by 6–h specific DEGs, expression of key genes encoding fructose-bisphosphate aldolase (VDAG_07429), 6-phosphate gluconate dehydrogenase (VDAG_04413), and transaldolase (VDAG_02628) was downregulated (Appendix A, Sheet 3). The common DEGs at 2–h and 6–h were enriched in the glycolysis/gluconeogenesis and pyruvate metabolism pathways, and the two rate-limiting enzymes of the glycolysis pathway, 6-phosphate fructose kinase (VDAG_01887) and pyruvate kinase (VDAG_01206), as well as pyruvate dehydrogenase (VDAG_06356, VDAG_01642), dihydrolipoyl dehydrogenase (VDAG_09433), and malate dehydrogenase (VDAG_08259), were also significantly downregulated (Appendix A, Sheet 6). The downregulation of these genes suggested that the aerobic oxidation of sugar, the main source of acetyl-CoA, was inhibited, which in turn slowed down the main energy production processes, such as the TCA cycle and oxidative phosphorylation in fungi, greatly reduced cell productivity, and seriously affected synthesis of intracellular biological macromolecules with acetyl-CoA as the precursor.

Interestingly, we found that glyoxylate-cycle-rate-limiting enzyme genes, such as isocitrate lyase (VDAG_08615, VDAG_02768) and malate synthase (VDAG_01774), were upregulated in the glyoxylate and dicarboxylate metabolism pathway, which may be because cells convert fat into sugar and then oxidize it completely to increase cell productivity. The upregulated expression of fatty acid degradation-related genes (VDAG_03899acyl CoA dehydrogenase family protein, VDAG_08463, VDAG_01631, VDAG_08250, VDAG_09022) in the fatty acid metabolism pathway may also increase cell productivity and maintain cell survival as much as possible. In summary, we believe that C17 mycosubtilin could significantly inhibit energy production of fungal cells.

### 3.11. qRT–PCR Verification of Gene Expression

To verify the changes in gene expression in RNA sequence analysis, nine genes with different expression patterns in three groups of genes with significant differences were measured by qRT–PCR. Figure 8A–C represents the 2–h and 6–h common DEGs related to the cell wall (downregulated), D–F represents 2–h specific DEGs related to hydrolase activity (upregulated), and G–I represents 6–h specific DEGs related to translation (downregulated). Their expression patterns were consistent with the results of RNA-seq, which indicated that the data analysis was reliable.

## 4. Conclusions

In this study, the antifungal mechanism of cyclic lipopeptide C17 mycosubtilin against Vd 991 was investigated. We found that C17 mycosubtilin inhibited fungal growth by affecting synthesis of the cell membrane and cell wall, inhibiting fungal DNA replication and translation, blocking the cell cycle, destroying cell energy and substance metabolism, disrupting the redox process, and so on. In addition, C17 mycosubtilin treatment changed the morphology of conidia and hyphae of Vd 991 and damaged the cell membrane and mitochondria.

## 5. Future Recommendations

To date, research on the mechanism of lipopeptides has mostly involved in vitro experiments. Some articles have reported that iturin A interacts with sterols or ergosterols on target cell membranes to disrupt integrity of plasma membranes, resulting in membrane perforation and leakage of cell contents [23,26]. C17 mycosubtilin specifically interacts with cholesterol-containing artificial membranes and ergosterol-containing interfacial monolayers [32,33]. Some studies have even proven that bacillomycin L can bind to fungal DNA non-specifically in vitro [95]. However, there is no evidence to prove that lipopeptides, such as iturin A and C17 mycosubtilin, can enter cells, so it is not clear whether C17 mycosubtilin directly acts on mitochondria and affects replication, transcription, and translation of DNA. In follow-up research, research on whether lipopeptides can enter the membrane should be strengthened. Previous studies showed that iturin A can induce apoptotic death of fungi, such as *Phytophthora infestans* [55] and *Aspergillus carbonarius* [24]. However, in our study, C17 mycosubtilin induced Vd 991 cell death mainly through necrosis. Only a small number of spores (<5%) showed apoptotic death after C17 mycosubtilin treatment, and no obvious apoptotic characteristics, such as chromatin aggregation, cytoplasmic condensation, and apoptotic bodies, were observed under a transmission electron microscope. Previous studies have shown that different polar groups and fatty acid chain lengths of lipopeptide compounds will affect the activity and mode of action of lipopeptides [96,97]. Therefore, the effect of C17 mycosubtilin on apoptosis of Vd 991 needs to be further studied. In short, our study revealed the antifungal mechanism of C17 mycosubtilin on Vd 991, providing clues to the mechanism of action of lipopeptides and effective information for development of more effective antimicrobials.

## Figures and Tables

**Figure 1 biology-12-00513-f001:**
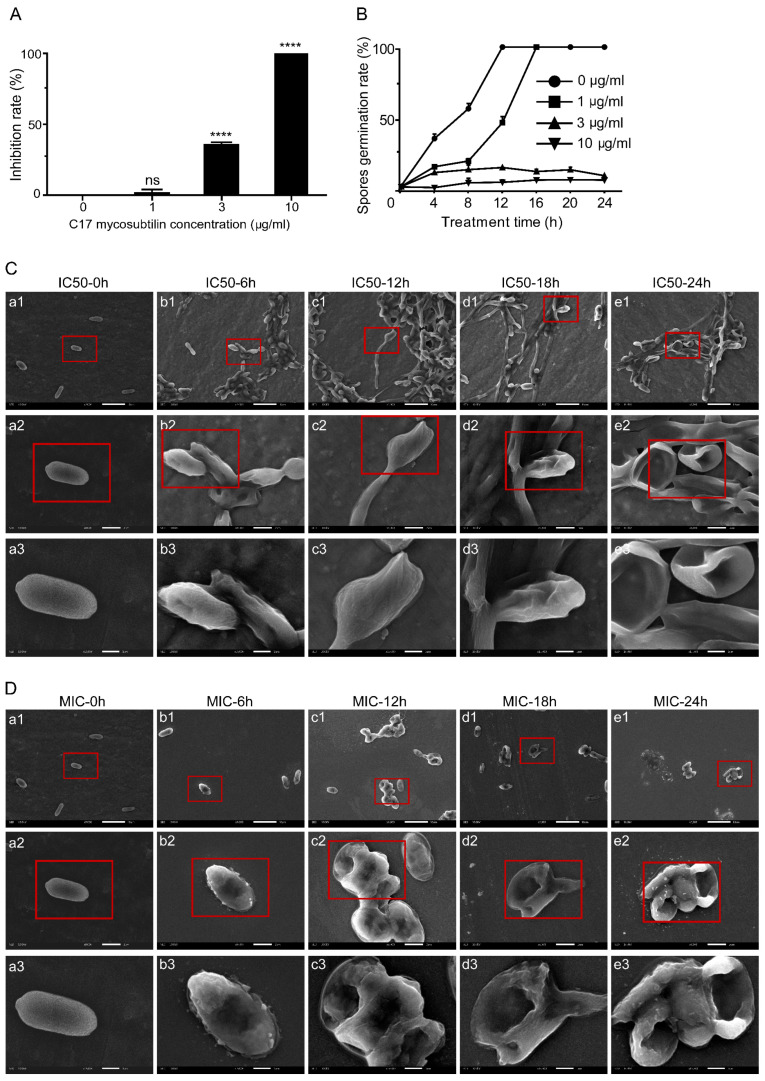
C17 mycosubtilin inhibits the growth and conidia germination of Vd 991 and affects its microscopic morphology. (**A**). The inhibition rate of different concentrations of C17 mycosubtilin on the growth of Vd 991. Data represent the average of three repetitions ± standard error (SD); statistical analyses were performed using one-way ANOVA. **** indicates extreme significance at *p* < 0.0001. ns: not significant. (**B**). Effects on spore germination of Vd 991 with C17 mycosubtilin. At least 200 spores were counted in each group, with 3 replicates. Data are shown as the mean ± SD. (**C**,**D**). SEM images of Vd 991 spores. Vd 991 was treated with C17 mycosubtilin at IC50 (**C**) and MIC (**D**) concentrations at different times. a, untreated Vd 991 spores (control); b1–e1, Vd 991 spores treated for 6 h, 12 h, 18 h, and 24 h, respectively. The red-framed area is the area displayed in the next row (b2–e2, and b3–e3, respectively).

**Figure 2 biology-12-00513-f002:**
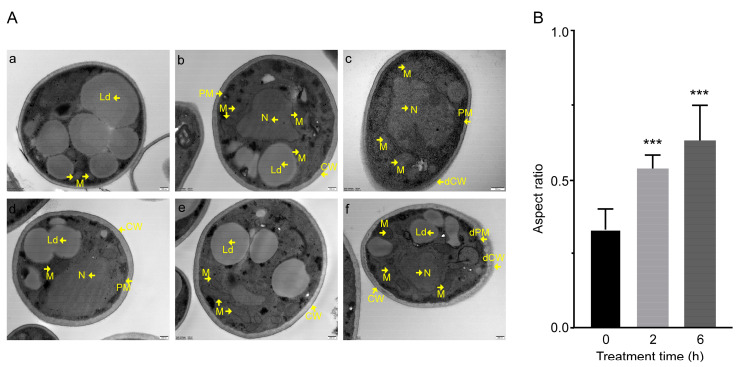
The effect of C17 mycosubtilin at the IC50 concentration on the morphology and structure of different organelles of Vd 991. (**A**). Transmission electron microscope (TEM) images of Vd 991 spores. a, d, untreated Vd 991 spores (control); b, e, Vd 991 spores treated with C17 mycosubtilin for 2 h at IC50 concentrations; c, f, Vd 991 spores treated with C17 mycosubtilin for 6 h at IC50 concentrations. M, mitochondria; N, nucleus; Ld, lipid droplets; CW, cell wall; dCW, damaged cell wall; PM, plasma membrane; dPM, damaged plasma membrane. (**B**). Mitochondrial aspect ratio chart. The aspect ratio of spore mitochondria treated with the IC50 concentration of C17 mycosubtilin for 0 h, 2 h, and 6 h was calculated. At least 30 mitochondria in each group were counted in each experiment, and the experiment was repeated 3 times. Statistical analyses were performed using one-way ANOVA. No less than 30 mitochondria per group were counted, and the values are the mean ± SD. *** indicates extreme significance at *p* < 0.001.

**Figure 3 biology-12-00513-f003:**
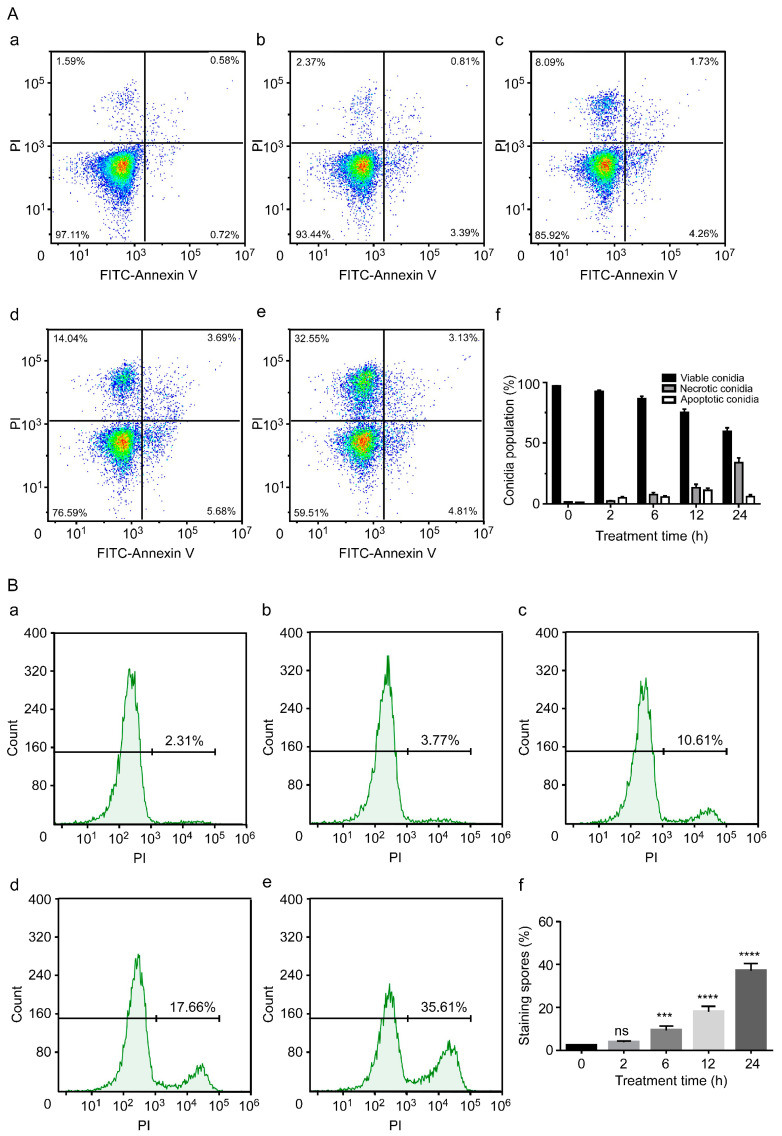
The conidial necrosis of Vd 991 was analysed by flow cytometry after treatment with C17 mycosubtilin. (**A**). The spore population of Vd 991 after treatment with C17 mycosubtilin at the IC50 concentration at different times by annexin V-FITC/PI double staining. a, Untreated Vd 991 spores (control); b–e, Vd 991 spores treated for 2 h, 6 h, 12 h, and 24 h, respectively; f, summary of spore population data. Each dot represents a cell, and the color represents the cell density (**B**). Vd 991 spores treated with C17 mycosubtilin at the IC50 concentration at different times by PI staining. a, Untreated Vd 991 spores (control); b–e, Vd 991 spores treated for 2 h, 6 h, 12 h, and 24 h, respectively; f, statistical diagram of spore staining rate at different treatment times. Statistical analysis in this figure is based on one-way ANOVA. Values are mean ± SD, where n ≥ 3; ns means no significant difference, *** means significant difference at *p* < 0.001, and **** means significant difference at *p* < 0.0001.

**Figure 4 biology-12-00513-f004:**
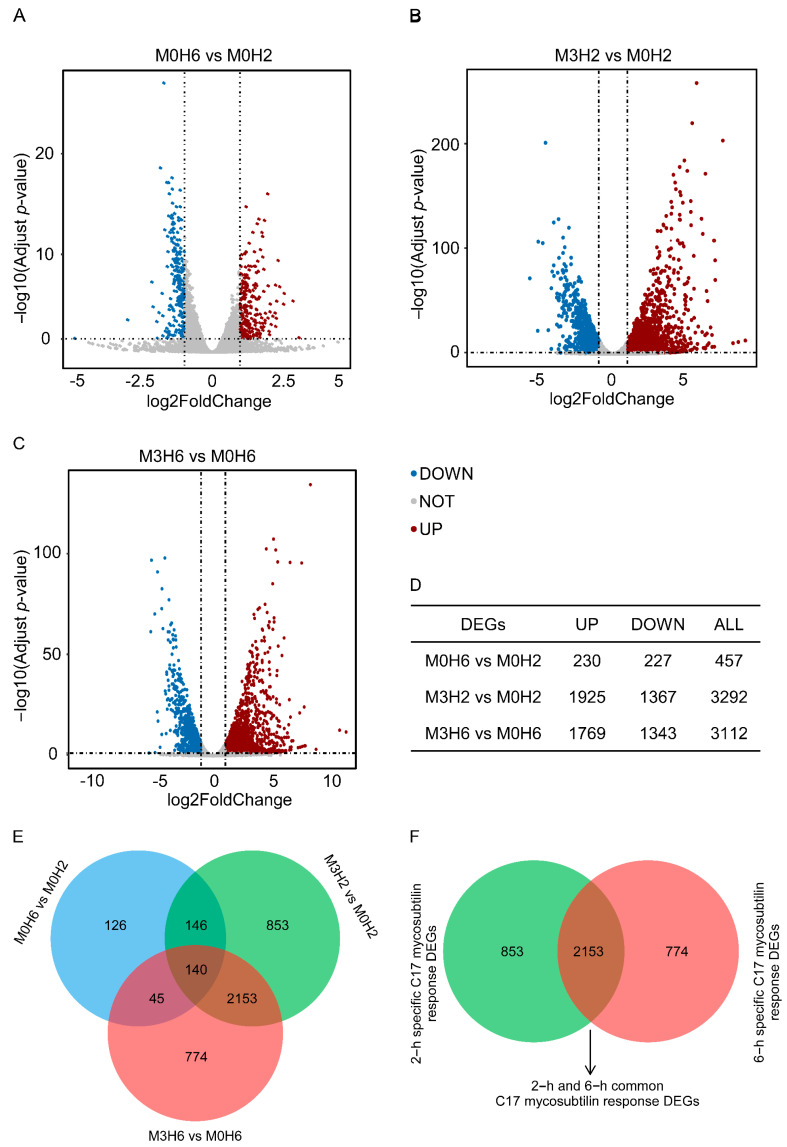
C17-mycosubtilin-responsible genes in Vd 991 spores after treatment at different times. (**A**–**C**) Volcano map of differentially expressed genes. Genes with a Padj (adjusted *p*-value) <0.05 and absolute fold change ≥2 were significantly DEGs. The red dots represent genes that are significantly upregulated, the blue dots represent genes that are significantly downregulated, and the grey dots represent genes that are not significantly differentiated. M0H6 and M0H2 indicated that the spores grew naturally in Czapek–Dox broth medium without C17 mycosubtilin treatment for 6 h and 2 h, respectively; M3H6 and M3H2 indicated that the spores grew in Czapek–Dox broth medium containing C17 mycosubtilin at the IC50 concentration for 6 h and 2 h, respectively. (**D**) Statistical results of different genes of each group. (**E**) Venn diagram of each group of differentially expressed genes. (**F**) Venn diagram after excluding differentially expressed genes produced in the natural growth state.

**Figure 5 biology-12-00513-f005:**
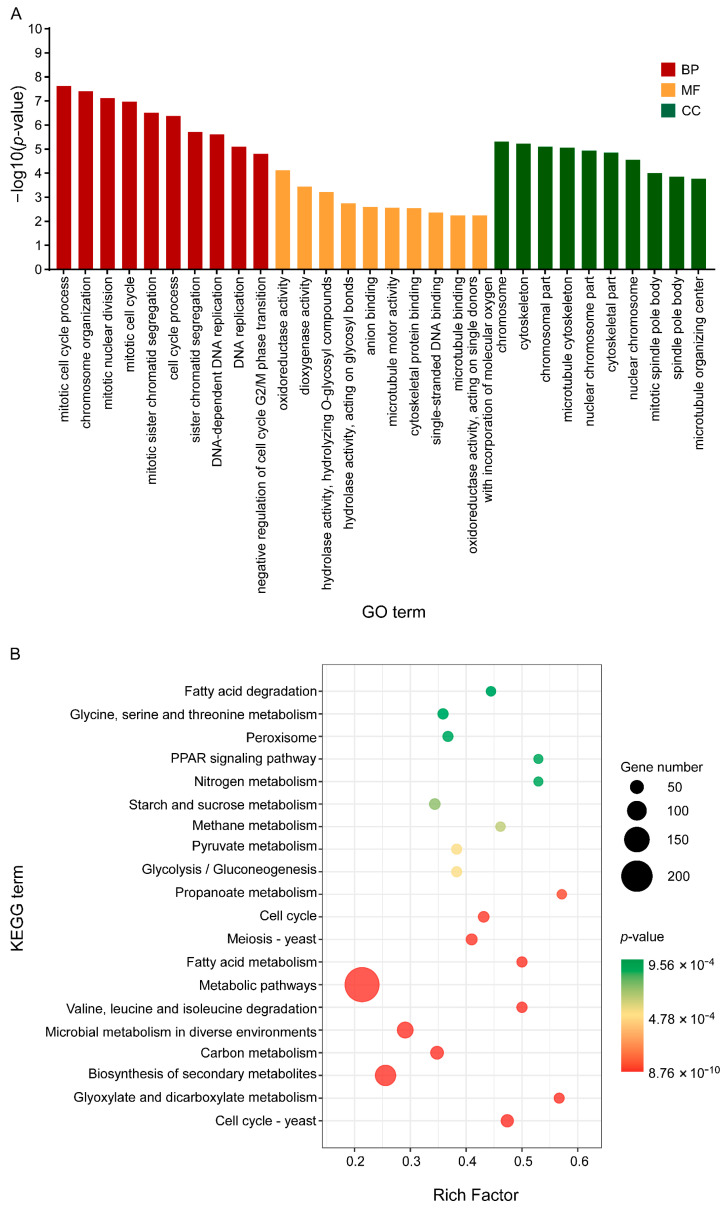
GO and KEGG enrichment map of the common C17-mycosubtilin-responsible genes with C17 mycosubtilin treatment at 2 and 6 h. (**A**). GO functional classification of the DEGs by *p*-value. The x-axis indicates different GO terms, and the y-axis indicates −log10 (corrected *p*-value) in each category. The top 10 enriched pathways of each category are shown in the figure. (**B**). Scatter plot of enriched KEGG pathways. The rich factor is the ratio of the number of genes that were differentially expressed to the total number of genes in a certain pathway. The top 20 enriched pathways are shown in the figure.

**Figure 6 biology-12-00513-f006:**
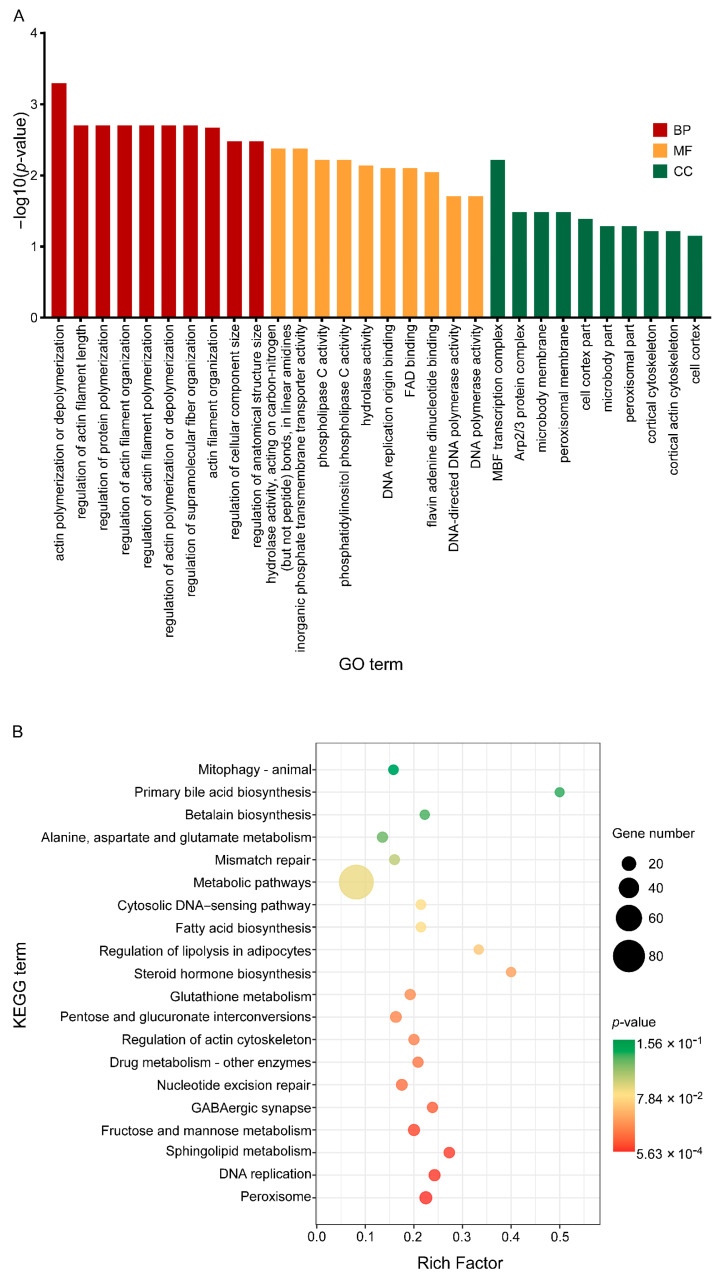
GO and KEGG enrichment map of the specific C17-mycosubtilin-responsible genes with C17 mycosubtilin treatment at 2 h. A. GO functional classification of the DEGs by *p*-value. The x-axis indicates different GO terms, and the y-axis indicates −log10 (corrected *p*-value) in each category. The top 10 enriched pathways of each category are shown in the figure. B. Scatter plot of enriched KEGG pathways. The rich factor is the ratio of the number of genes that were differentially expressed to the total number of genes in a certain pathway. The top 20 enriched pathways are shown in the figure.

**Figure 7 biology-12-00513-f007:**
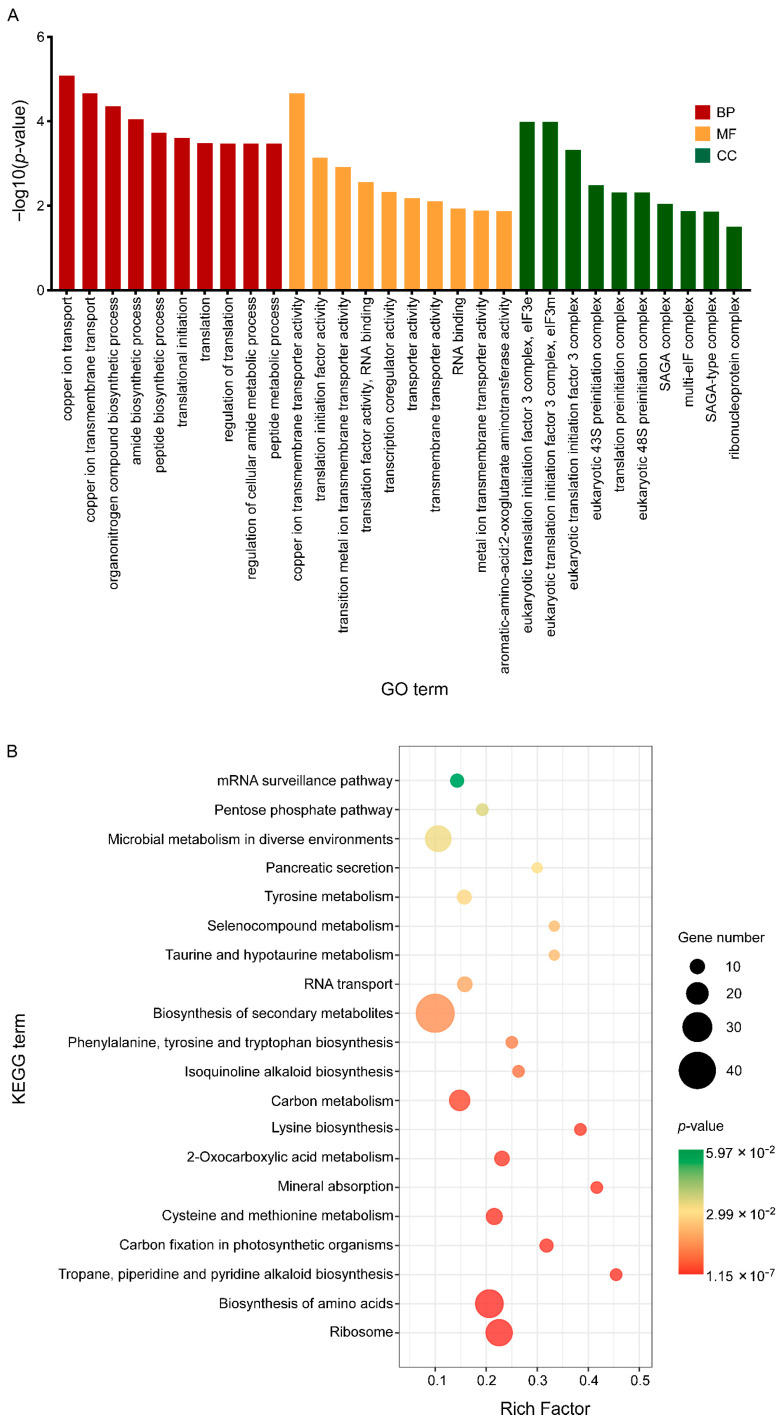
GO and KEGG enrichment map of the specific C17-mycosubtilin-responsible genes with C17 mycosubtilin treatment at 6 h. (**A**). GO functional classification of the DEGs by *p*-value. The x-axis indicates different GO terms, and the y-axis indicates −log10 (corrected *p*-value) in each category. The top 10 enriched pathways of each category are shown in the figure. (**B**). Scatter plot of enriched KEGG pathways. The rich factor is the ratio of the number of genes that were differentially expressed to the total number of genes in a certain pathway. The top 20 enriched pathways are shown in the figure.

**Figure 8 biology-12-00513-f008:**
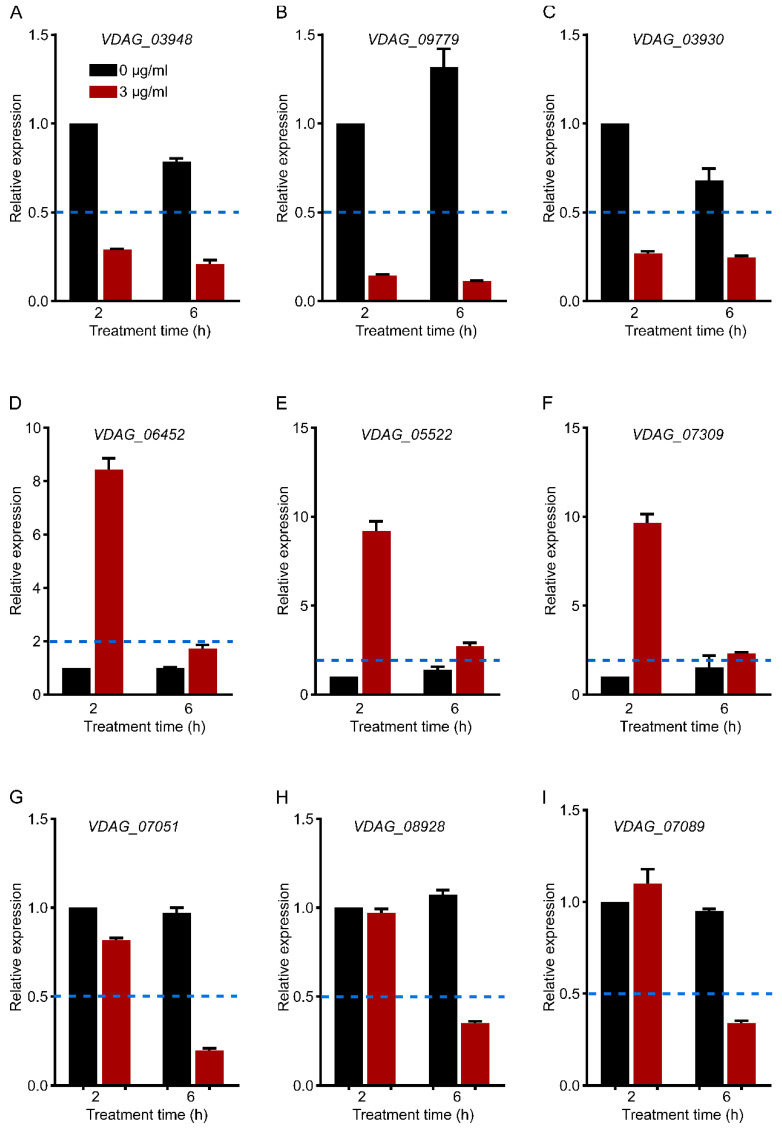
qRT–PCR verification of gene expression (**A**–**C**). Cell-wall-related genes (2–h and 6–h common DEGs). (**D**–**F**). Hydrolase-activity-related genes (2–h specific DEGs). (**G**–**I**). Translation-related genes (6–h specific DEGs). Each gene expression level was normalized to that of *Vdβ-tubulin*, and the relative expression of each gene at 2–h without C17 mycosubtilin treatment was set at 1.0. Values are means ± SDs of three independent experiments. If the relative expression level is more than two times different from that of the control group (<0.5 or >2), it is considered to be significantly upregulated or downregulated. The blue dotted line represents a relative expression level of 0.5 or 2.

## Data Availability

The RNA-seq data were deposited in the National Center for Biotechnology Information (NCBI) Sequence Read Archive (SRA) (http://www.ncbi.nlm.nih.gov/bioproject/868877, accessed on 31 August 2022), under the accession number PRJNA868877. All data generated or analyzed during this study are included in this article and its Appendix A.

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
