# Peer review of "Transcriptome Analysis Reveals That C17 Mycosubtilin Antagonizes Verticillium dahliae by Interfering with Multiple Functional Pathways of Fungi"

_biology, 2023, doi:10.3390/biology12040513_

Round 1
Reviewer 1 Report
The study proposed by Zhang et al. presents the effect of mycosubtiline C17 on a phytopathogenic fungus Verticillium dahliae. In this work, the authors investigate in depth the impact of the molecule on spores and hyphae but also on the metabolic function of the fungus through a transcriptomic study.
The manuscript is very well written and the methodologies and techniques used are appropriate. The results are original and bring new information extremely interesting for the scientific community working on the subject of lipopeptides.
I suggest some minor additions and modifications to be seen directly in the attached file

Reviewer 2 Report
Verticillium dahliae (Vd) is one of the most important soil-borne fungi which could cause about 400 crop species wilt. Until now, the main research was focused on the isolation and identification of the antimicrobials of Vd. In this study, based on previous research of C17 mycosubtilin from the Bacillus subtilis J15 (BS J15) strain, its antagonistic mechanism was further investigated. As a whole, the data is solid and the conclusions are proper. However, some details should be revised. Below some minor suggestions are listed:
1. The special words should be in abbreviation style when it appears again. E.g. line 58 Verticillium dahliae (Vd), line 85 Fusarium oxysporum, line 86 Fusarium graminearum, line 95 Fusarium graminearumschw, line 116 Bacillus subtilis, line 288…check the whole manuscript.
2. line 252 and line 253-254, the description of qRT‒PCR is different and it should be used qRT‒PCR in line 253.
3. line 270, it should be 3.
4. in Fig. 1A, Fig. 8, statistical analysis of the data should be performed by one-way ANOVA.
5. line 806-807, “provided clues to the mechanism of action of lipopeptides, and provided effective information for the development of more effective antimicrobials”. The expression should be revised. Two “provided” and “effective”.
6. describe the M0H6, M0H2, M3H6 in manuscript. E.g. line 369 at IC50 concentrations for 2 h (M0H2) and 6 h (M0H6) were studied.
